# Water Quality Prediction Based on Machine Learning and Comprehensive Weighting Methods

**DOI:** 10.3390/e25081186

**Published:** 2023-08-09

**Authors:** Xianhe Wang, Ying Li, Qian Qiao, Adriano Tavares, Yanchun Liang

**Affiliations:** 1School of Applied Chemistry and Materials, Zhuhai College of Science and Technology, Zhuhai 519041, China; wxh@zcst.edu.cn (X.W.); liying@zcst.edu.cn (Y.L.);; 2Department of Industrial Electronics, School of Engineering, University of Minho, 4704-553 Braga, Portugal; 3School of Computer Science, Zhuhai College of Science and Technology, Zhuhai 519041, China; 4Key Laboratory of Symbol Computation and Knowledge Engineering of the Ministry of Education, College of Computer Science and Technology, Jilin University, 2699 Qianjin Street, Changchun 130012, China

**Keywords:** water quality prediction, comprehensive weight-based approach, feature selection, machine learning, LSTM

## Abstract

In the context of escalating global environmental concerns, the importance of preserving water resources and upholding ecological equilibrium has become increasingly apparent. As a result, the monitoring and prediction of water quality have emerged as vital tasks in achieving these objectives. However, ensuring the accuracy and dependability of water quality prediction has proven to be a challenging endeavor. To address this issue, this study proposes a comprehensive weight-based approach that combines entropy weighting with the Pearson correlation coefficient to select crucial features in water quality prediction. This approach effectively considers both feature correlation and information content, avoiding excessive reliance on a single criterion for feature selection. Through the utilization of this comprehensive approach, a comprehensive evaluation of the contribution and importance of the features was achieved, thereby minimizing subjective bias and uncertainty. By striking a balance among various factors, features with stronger correlation and greater information content can be selected, leading to improved accuracy and robustness in the feature-selection process. Furthermore, this study explored several machine learning models for water quality prediction, including Support Vector Machines (SVMs), Multilayer Perceptron (MLP), Random Forest (RF), XGBoost, and Long Short-Term Memory (LSTM). SVM exhibited commendable performance in predicting Dissolved Oxygen (DO), showcasing excellent generalization capabilities and high prediction accuracy. MLP demonstrated its strength in nonlinear modeling and performed well in predicting multiple water quality parameters. Conversely, the RF and XGBoost models exhibited relatively inferior performance in water quality prediction. In contrast, the LSTM model, a recurrent neural network specialized in processing time series data, demonstrated exceptional abilities in water quality prediction. It effectively captured the dynamic patterns present in time series data, offering stable and accurate predictions for various water quality parameters.

## 1. Introduction

With the increasing human activities associated with industrialization and urbanization development, the water quality of coastal rivers is facing escalating and severe threats and degradation [1]. Coastal rivers play a critical role in connecting land and ocean, and the water quality directly impacts the well-being and sustainable development of coastal ecosystems. Therefore, it is imperative to recognize and address the pressing issue of deteriorating water quality in coastal rivers [2]. To protect river water quality, maintain the integrity of coastal ecosystems, and ensure sustainable human development, effective management and protection measures must be implemented [3]. Advanced technological means and scientific methods should be employed to strengthen water quality monitoring, early warning systems, and governance capabilities [4].

Traditional river water quality monitoring and warning technology relies on theoretical models that encompass physical, chemical, and biological processes [5]. These models describe and predict changes in water quality parameters by establishing mathematical equations. Mechanism models typically consider factors such as water flow velocity, flow rate, water quality parameters, as well as the transport and transformation of pollutants [6]. Common mechanism models include hydrodynamic models, water quality models, and ecological models. Traditional water-quality-monitoring and early warning technologies based on mechanism models offer certain advantages [7]. They are grounded in a profound understanding of hydrology, hydrodynamics, water quality, and ecological processes, thereby exhibiting high interpretability and reliability [8]. These models enable quantitative prediction and analysis of water quality variations, facilitating the assessment of the health of the water environment and the formulation of strategies to improve water quality [9]. However, traditional mechanism models also possess certain limitations. Firstly, they typically require extensive input data and parameters, including flow rate, rainfall, sediment characteristics, etc., which entail complex data acquisition and processing [10]. Secondly, establishing and calibrating the model necessitate deep professional knowledge and a substantial volume of measured data, demanding high technical expertise [11]. Moreover, the representation of complex water environments and ecosystems by mechanism models may involve simplifications and idealizations that fail to fully capture the complexity of real-world situations.

In recent years, there has been extensive research and applications of machine-learning-based technology for predicting river water quality [12]. In the context of river water quality prediction, machine learning utilizes a large amount of historical water quality data to construct accurate prediction models and enable early warning. This technology offers several advantages [13]. Firstly, it facilitates real-time and continuous monitoring and prediction of water quality, enhancing the responsiveness and effectiveness of water quality management. Secondly, machine learning models can automatically learn and adapt to the complex relationships within water quality data, resulting in more-accurate predictions [14]. Additionally, these models can incorporate other environmental factors and meteorological data, thereby improving the accuracy and reliability of water quality prediction. However, machine learning methods also face challenges and limitations in predicting river water quality. Issues such as data quality and missing data can impact the model’s performance [15]. Moreover, training and parameter selection require a certain level of professional knowledge and experience. Additionally, the interpretability of the model is relatively low, making it difficult to interpret the predicted results. Therefore, further research and improvement are necessary to enhance the effectiveness and reliability of machine learning in river water quality prediction.

The entropy weighting method is an information-theory-based approach used to evaluate the information content and importance of features [16]. By computing the entropy value of features, the purity and discriminability of the features can be measured [17]. By combining the Pearson correlation coefficient with the entropy weighting method, the correlation and information content of the features can be comprehensively considered, avoiding over-reliance on a single criterion for feature selection. This approach enables a more-comprehensive evaluation of the contribution and importance of features, reducing subjectivity and uncertainty. By balancing different factors, features with higher correlation and greater information content can be selected, thereby improving the accuracy and stability of feature selection. Both the Pearson correlation coefficient and the entropy weighting method are relatively simple and intuitive approaches, making them easy to understand and interpret [18]. By integrating them into feature selection in machine learning, feature-selection results with higher interpretability and practicality can be obtained. This enhances the transparency and reliability of the feature-selection process, helping decision-makers understand the importance and contribution of features while improving the performance and interpretability of the model [19].

The objective of this study was to utilize machine learning techniques, in conjunction with the entropy weight method and Pearson correlation coefficient method as feature-selection methods, to achieve high-precision prediction of major water quality indicators, such as Dissolved Oxygen (DO), Ammonia Nitrogen (NH${}_{3}$-N), Total Phosphorus (TP), and Total Nitrogen (TN). The models considered in this study encompass Long Short-Term Memory (LSTM), Support Vector Machine (SVM), Multilayer Perceptron (MLP), Random Forest (RF), and XGBoost. LSTM, as a variant of the Recurrent Neural Network (RNN) suitable for processing time series data, exhibited superior performance in predicting water quality changes. By inputting historical water quality data as time series, the LSTM model can be used to effectively capture long-term dependencies and accurately forecast future trends in water quality changes. Furthermore, other machine learning models offer distinct advantages and applicability in water quality prediction, enabling the selection of appropriate models based on specific requirements.

## 2. Materials and Methods

### 2.1. Data Acquisition

The data used for this research were sourced from the China Environmental Monitoring General Station, specifically from the monitoring point at Shijiaoju Section in the Pearl River Basin. The dataset comprises water quality measurements collected at four-hour intervals, covering the time period from 8 November 2020 to 28 February 2023. In total, there are 5058 samples in this dataset, encompassing 9 water quality parameters: Ammonia Nitrogen (NH${}_{3}$-N), water Temperature (Temp), pH, Dissolved Oxygen (DO), the permanganate index (KMnO${}_{4}$, Total Phosphorus (TP), Total Nitrogen (TN), Conductivity (Cond), and Turbidity (Turb).

### 2.2. Data Preprocessing

Data preprocessing is a vital step in machine learning, encompassing various tasks such as handling outliers, missing values, and data normalization [20]. For this study, historical monitoring data were collected, including water quality indicators such as temperature, pH, the potassium permanganate index, dissolved oxygen, ammonia nitrogen, total phosphorus, total nitrogen, turbidity, and conductivity. Firstly, outlier detection was performed on the data. Outliers can arise due to sensor malfunctions, human errors, or other factors, resulting in abnormal data points. Statistical analysis is commonly employed for outlier detection, involving calculations of the mean and standard deviation to identify values significantly deviating from the mean. If outliers are detected, they can be treated as missing values or corrected based on the specific circumstances [21]. Next, the focus was placed on addressing missing values in the data. Linear interpolation was utilized in this study to fill in the missing values. Linear interpolation estimates the missing values by considering the linear relationship between known data points. Specifically, for time series data, the observed values of the preceding and succeeding time points are used to perform linear interpolation and estimate the missing values. Suppose we want to estimate the missing value *x* between the known data points $x_{1}$ and $x_{2}$, corresponding to observed values $y_{1}$ and $y_{2}$, respectively. The linear interpolation formula for estimating the value y is as follows:(1)$$y=y_{1}+\left(x-x_{1}\right)\frac{y_{2}-y_{1}}{x_{2}-x_{1}}$$

Here, (*x*
$-\;x_{1}$) represents the offset of *x* relative to $x_{1}$ and (($y_{2}$ − $y_{1}$)/($x_{2}$ − $x_{1}$)) represents the slope from $x_{1}$ to $x_{2}$. By multiplying the offset by the slope and adding it to $y_{1}$, the missing value y can be estimated [22].

The advantages of linear interpolation include its simplicity, ease of use, and the ability to produce reasonably accurate estimation results in certain cases [23]. However, linear interpolation also has limitations. Firstly, it assumes a linear relationship between data points, which may not hold true in all situations. Secondly, it requires a high density of data points, and sparse or unevenly distributed data may lead to inaccurate estimates. Additionally, linear interpolation cannot capture nonlinear trends or special patterns in the data [24]. Therefore, when applying linear interpolation, it is crucial to assess the characteristics and validity of the data within the specific context and consider the suitability of alternative interpolation methods [25].

Lastly, data normalization will be performed to eliminate dimensional differences among different water quality indicators [26]. The chosen method was min–max normalization, which linearly transforms the data to a specific range, typically [0, 1] or [−1, 1], ensuring that feature variables have similar scales. Min–max normalization can be calculated using the following formula:(2)$$X_{N}=\frac{X-X_{min}}{X_{max}-X_{min}}$$

Here, $X_{N}$ represents the normalized value, *X* represents the original value, $X_{min}$ represents the minimum value and $X_{max}$ represents the maximum value. This formula maps the original data to a range between 0 and 1.

By following these data preprocessing steps, we can obtain cleaned and prepared data suitable for the subsequent feature selection, training, and prediction of machine learning models. This process will enhance the accuracy and stability of the models and provide a reliable foundation for predicting river water quality.

### 2.3. Feature Variable Selection

#### 2.3.1. Entropy Weighting Method

The entropy weight method is a technique employed to determine the weights of multiple indicators. It utilizes the concept of entropy to measure the uncertainty or diversity of indicators by computing their information entropy [27]. This method finds widespread application in multi-indicator decision-making, evaluation, and ranking, aiding in addressing challenges associated with trade-offs and optimization among multiple indicators [28]. The steps involved in calculating weights using the entropy weighting method based on information entropy are as follows:Calculate information entropy: Compute the information entropy for each indicator. The information entropy quantifies the uncertainty or diversity of an indicator, with higher entropy values indicating greater diversity. The calculation formula for the information entropy is as follows:
(3)$$H\left(X\right)=-\sum \left(P_{i}\times \log_{2}P_{i}\right)$$Here, $H\left(X\right)$ represents the information entropy of the indicator and $P_{i}$ represents the normalized value of the indicator.Calculate information weight: Determine the information weight for each indicator based on its information entropy. The calculation formula for the information weight is as follows:
(4)$$W_{i}=\left(1-H\left(X_{i}\right)\right)$$Here, $W_{i}$ denotes the information weight of indicator $X_{i}$ and $H(X_{i})$ represents the information entropy of indicator $X_{i}$.

The information weight reflects the level of information contained within an indicator. A higher information weight signifies a greater impact of the indicator on the decision outcome. Hence, information weights can be utilized to assess the significance of indicators and their contributions to the decision-making process. By calculating the information entropy and information weights for each indicator, a weight vector can be derived for subsequent tasks such as multi-indicator decision-making, evaluation, or optimization [29]. Incorporating information weights assists decision-makers in making informed trade-offs and selections among indicators, thus enhancing the accuracy and credibility of decisions. However, it is essential to recognize that information weights solely consider the diversity and uncertainty of indicators, disregarding their interrelationships. Therefore, in practical applications, it is crucial to consider additional methods or domain knowledge to comprehensively assess the indicators [30].

According to the analysis presented in Table 1 and Figure 1, several variables stand out with relatively higher weights. Specifically, NH${}_{3}$-N, DO, KMnO${}_{4}$, TP, Cond, and Turb exhibit higher weights compared to other variables. Among these, NH${}_{3}$-N, Cond, and Turb emerge as particularly influential indicators, indicating their significance and stronger influence on the decision outcome. Conversely, Temp, pH, and TN display relatively lower weights, implying their diminished importance and weaker impact on the decision outcome.

#### 2.3.2. Pearson Correlation Coefficient Method

The Pearson correlation coefficient is a statistical measure utilized to evaluate the linear correlation between two continuous variables [31]. It provides information about the strength and direction of the linear relationship between the variables, making it a commonly employed method for feature variable selection and evaluation [32]. The coefficient, denoted as “r”, ranges from −1 to 1. A value of r = 1 indicates a perfect positive linear relationship, while r = −1 indicates a perfect negative linear relationship. A value of r = 0 suggests no linear relationship, indicating no correlation between the variables. The calculation formula is as follows:(5)$$R=\frac{\sum \left(X_{i}-X_{mean}\right)\left(Y_{i}-Y_{mean}\right)}{NX_{std}Y_{std}}$$
Here, *R* represents the Pearson correlation coefficient, $X_{i}$ and $Y_{i}$ denote the values of the variables in the observation matrix, $X_{mean}$ and $Y_{mean}$ represent the means of the variables, $X_{std}$ and $Y_{std}$ represent the standard deviations of the variables, and *N* denotes the number of observations in the sample.

According to Figure 2, significant correlation coefficients were observed between NH${}_{3}$-N and pH, DO, KMnO${}_{4}$, TP, and TN, indicating a substantial relationship. DO exhibited significant correlation coefficients with NH${}_{3}$-N, temperature, pH, KMnO${}_{4}$, TP, conductivity, and turbidity, indicating significant relationships. Similarly, TP showed significant correlation coefficients with NH${}_{3}$-N, temperature, pH, DO, KMnO${}_{4}$, TN, conductivity, and turbidity, indicating significant relationships. Likewise, TN exhibited significant correlation coefficients with NH${}_{3}$-N, temperature, pH, KMnO${}_{4}$, TP, conductivity, and turbidity, indicating significant relationships.

#### 2.3.3. Comprehensive Weight Method

The comprehensive weight method is an approach for selecting feature variables that combines the Pearson correlation coefficient method and the entropy weight method. It aims to evaluate and select feature variables by calculating their comprehensive weights, which are obtained by multiplying the Pearson correlation coefficient value of each feature variable with its corresponding information weight. The formula for calculating the comprehensive weight is as follows:(6)$$V_{CW}=V_{PCC}\times V_{IW}$$
Here, $V_{CW}$ represents the comprehensive weight, $V_{PCC}$ represents the Pearson correlation coefficient, and $V_{IW}$ represents the information weight. A higher comprehensive weight indicates a greater importance and relevance of the feature variable in predicting the target variable.

The comprehensive weight method offers the advantage of considering both linear relationships and importance factors, resulting in a more-comprehensive evaluation and selection of feature variables. This, in turn, improves the accuracy and stability of the feature-selection process. Based on the comprehensive weight calculation results presented in Table 2 and Figure 3, the following input variables were selected for predicting the respective target variables:Dissolved Oxygen (DO) prediction: DO, NH${}_{3}$-N, Temp, pH, KMnO${}_{4}$, TP, Cond, and Turb;Ammonia Nitrogen (NH${}_{3}$-N) prediction: NH${}_{3}$-N, DO, KMnO${}_{4}$, TP, and TN;Total Nitrogen (TN) prediction: TN, NH${}_{3}$-N, KMnO${}_{4}$, TP, Cond, and Turb;Total Phosphorus (TP) prediction: TP, NH${}_{3}$-N, Temp, DO, KMnO${}_{4}$, TN, Cond, and Turb.

These selected input variables were determined based on their respective comprehensive weights, which consider both the Pearson correlation coefficient and information weight. By including these variables in the prediction models, it is expected to enhance the accuracy and reliability of the predictions for Dissolved Oxygen (DO), Ammonia Nitrogen (NH${}_{3}$-N), Total Nitrogen (TN), and Total Phosphorus (TP).

### 2.4. Models

#### 2.4.1. Support Vector Machine

Support Vector Machine (SVM) is a popular supervised learning algorithm utilized for classification and regression tasks. Its objective is to discover an optimal hyperplane that effectively separates different classes of samples while maximizing the margin between them, thereby achieving robust generalization performance [33]. SVM achieves this by mapping the samples into a high-dimensional feature space and identifying the hyperplane that maximizes the margin within this space. This hyperplane is defined as the one with the greatest distance to the nearest samples of different classes, referred to as support vectors, which play a crucial role in determining the hyperplane’s position.

SVM demonstrates high accuracy and generalization performance, particularly for small-sized datasets. It exhibits robustness against noise and outliers and is well-suited for handling high-dimensional data. Moreover, the decision function of SVM is based on the support vectors, which provide valuable insights into the data distribution and decision boundary, thus offering interpretability to some extent. However, SVM also has certain limitations. It can be computationally slow when applied to large-scale datasets, and its performance may degrade on high-dimensional data or when dealing with imbalanced classes. Additionally, the selection of appropriate kernel functions and tuning of the related parameters are important considerations when utilizing SVM [34].

#### 2.4.2. Multilayer Perceptron

Multilayer Perceptron (MLP) is a neural network with an input layer, multiple hidden layers, and an output layer. It uses weighted connections and nonlinear activation functions to process data. The network is trained using the backpropagation algorithm, updating weights to minimize prediction errors. MLP excels in modeling complex patterns and can be adjusted to fit different task complexities [35]. However, training and prediction times may be longer for large-scale or high-dimensional data. The model’s performance depends on factors such as activation functions, the architecture, and the hyperparameters [36]. Multiple metrics should be used for evaluation, considering the model structure, feature selection, and data distribution. Enhancements can be made through adjustments, optimization, additional features, or alternative algorithms [37].

#### 2.4.3. Random Forest

Random Forest is an ensemble learning method that constructs multiple weak learners based on decision trees. It combines the predictions of individual trees through voting or averaging to make the final predictions. In each node of the decision tree, Random Forest considers only a random subset of features for splitting. This selective feature consideration reduces the correlation between trees, leading to increased model diversity. To create diverse decision trees, multiple training sets are generated using bootstrap sampling. This process involves randomly selecting samples with replacement from the original training set, enabling the training of different decision trees [17]. The utilization of bootstrap sampling enhances model diversity and mitigates overfitting. Random Forest generates predictions by aggregating the collective decisions of multiple decision trees. In classification tasks, the prediction is determined by the majority class obtained through voting, while in regression tasks, the average prediction of the multiple decision trees is used. Random Forest models are highly proficient in handling high-dimensional and large-scale data, demonstrating their suitability for complex nonlinear relationships [33]. Furthermore, they exhibit robustness in the presence of missing values and outliers.

#### 2.4.4. Extreme Gradient Boosting

Extreme Gradient Boosting (XGBoost) is an algorithm developed based on the gradient boosting decision trees approach. It enhances model accuracy and efficiency by incorporating regularization techniques and parallel computing. XGBoost leverages the gradient boosting algorithm, which iteratively trains a sequence of decision trees to progressively enhance the predictive model’s performance. Each tree is trained to rectify the prediction errors made by the preceding tree, gradually aligning with the negative gradient of the objective function. To address the risk of overfitting, XGBoost employs various regularization techniques, including L1 and L2 regularization. Additionally, constraints on tree depth and leaf weights are applied to manage the model’s complexity and prevent overfitting. These measures collectively contribute to improving the overall performance and generalization capability of the XGBoost algorithm [38].

#### 2.4.5. Long Short-Term Memory

Long Short-Term Memory (LSTM) is a specialized variant of Recurrent Neural Networks (RNNs) that excels in processing time series data. Unlike conventional RNNs, LSTM incorporates gating mechanisms that effectively capture and retain long-term dependencies [39]. The core of an LSTM network comprises three essential gate units: the forget gate, the input gate, and the output gate. These gate units regulate the flow and manipulation of information through learnable weights, thereby controlling the input, output, and memory processes. By selectively forgetting, updating, and outputting information, LSTM enables the model to effectively retain and utilize long-term data information, effectively addressing the challenge of long-term dependencies encountered in traditional RNNs. The gating mechanisms employed by LSTM also address the issues of vanishing and exploding gradients, ensuring smooth gradient propagation over extended time intervals [40]. LSTM exhibits versatility in handling diverse input and output types, including univariate and multivariate time series data, as well as text data. It offers flexibility in adjusting input and output dimensions and possesses strong representational capabilities [12]. LSTM has achieved remarkable success in various domains such as natural language processing, speech recognition, machine translation, and time series prediction. Consequently, LSTM finds widespread application in modeling and prediction tasks across a wide range of practical problems.

#### 2.4.6. Grid Search (GridSearchCV)

In this study, the Grid Search technique (GridSearchCV) was utilized to fine-tune the parameters of the machine learning model and identify the optimal combination of parameters, thereby enhancing the model’s performance and predictive capability. Grid search systematically explores all potential parameter combinations within the defined parameter ranges, extensively investigating the parameter space to determine the best configuration [41]. In practical applications, machine learning models often possess adjustable parameters such as the learning rate, regularization parameter, and tree depth, which significantly impact model performance. By employing grid search, we can methodically evaluate the performance of diverse parameter combinations and identify the optimal set of parameters to achieve the best model performance. The primary advantage of this approach lies in its comprehensiveness and intuitive nature. It eliminates the need for intricate mathematical derivations or optimization algorithms; instead, it involves specifying the parameter value ranges and exhaustively iterating through all feasible parameter combinations. Consequently, grid search is straightforward to understand and implement, providing interpretable results that facilitate a clear understanding of how different parameter combinations affect the model’s performance [42].

### 2.5. Model Evaluation

Model evaluation refers to assessing the performance of a trained model to understand how well it performs on unseen data. The following Table 3 provides the model evaluation metrics used in this study.

The evaluation metrics, the MSE and RMSE, are considered favorable when they have smaller values, while the NSE and $R^{2}$ score should approach 1. These metrics provide insights into the predictive performance of the model and facilitate the comparison of different models to select the most-suitable one [43].

Using GridSearchCV, we conducted a grid search to identify the optimal model structures and parameter combinations for each model. We also integrated the input variable sets selected by the comprehensive weight method to make separate predictions for Dissolved Oxygen (DO), Ammonia Nitrogen (NH${}_{3}$-N), Total Nitrogen (TN), and Total Phosphorus (TP) in the surface water. Moreover, we divided the dataset into a training set and a test set. The first 4552 samples were allocated to the training set, while the remaining 506 samples were designated as the test set. The training set was used to train the models, and the test set was employed to evaluate the model’s performance and assess its effectiveness in making predictions.

To enhance the stability and reliability of predictions by mitigating the impact of random factors, the model prediction experiments were conducted in parallel for 10 sets. The final evaluation result for the model’s prediction parameter was obtained by calculating the average of the $R^{2}$ values, MSE values, RMSE values, and NSE values from these 10 parallel experiments. This approach allows for a comprehensive assessment of the model’s performance across diverse experiments, reducing the potential influence of random errors associated with a single experiment [44]. Running multiple experiments yields a larger set of data points, which enhances the statistical significance of the evaluation results and provides a more-comprehensive and -accurate evaluation of the model’s performance.

## 3. Results and Discussion

### 3.1. Data Statistics

Based on the analysis of Table 4 and Figure 4, insights can be obtained regarding the distribution of the data for each variable. Notably, Dissolved Oxygen (DO) and Ammonia Nitrogen (NH${}_{3}$-N) exhibited substantial fluctuations throughout the year, while Total Nitrogen (TN) and Total Phosphorus (TP) demonstrated relatively smaller fluctuations. The standard deviations for Dissolved Oxygen (DO), Ammonia Nitrogen (NH${}_{3}$-N), and Total Nitrogen (TN) were calculated as 3.642, 0.464, and 0.672, respectively, indicating a higher level of data variability associated with these variables. Conversely, the standard deviation for Total Phosphorus (TP) was determined to be 0.047, suggesting a lower level of data variability.

Dissolved Oxygen (DO) represents the amount of oxygen dissolved in water and is influenced by various factors, with temperature being one of the primary influences. Generally, as temperature increases, the concentration of dissolved oxygen decreases. This relationship implies that higher temperatures lead to lower dissolved oxygen concentrations, while lower temperatures result in higher dissolved oxygen concentrations. The summer season, typically occurring from June to September in most regions, is associated with elevated temperatures. Consequently, water temperature rises during this period, leading to a decrease in dissolved oxygen concentration and lower values of dissolved oxygen. Conversely, in January and December, lower temperatures prevail, causing the water temperature to be relatively colder. As a result, the dissolved oxygen concentration increases, yielding higher values of dissolved oxygen. Although other factors such as oxygen supply, environmental conditions in the water (e.g., plant growth, water body mixing), and meteorological changes can influence the dissolved oxygen concentration, temperature remains the primary factor among them.

Ammonia Nitrogen (NH${}_{3}$-N) exhibits relatively higher concentrations from June to August each year due to several reasons. Firstly, the summer season corresponds to the peak period of biological activity in water, involving microorganisms, algae, and bacteria. These organisms absorb nutrients, including ammonia compounds, from the water for growth and metabolism, contributing to the release of ammonia nitrogen. Therefore, increased biological activity during summer results in higher concentrations of ammonia nitrogen. Secondly, higher air temperatures during summer lead to elevated water temperatures. Ammonia nitrogen’s solubility is positively correlated with water temperature, meaning that higher temperatures facilitate the dissociation of ammonia nitrogen molecules from solids or organic matter, increasing its solubility in water. Additionally, the concentration of ammonia nitrogen can be influenced by factors such as sediment release, agricultural and urban discharges, rainfall, and flow variations.

The concentration of Total Phosphorus (TP) shows relatively small and stable fluctuations throughout the year due to several factors. Stable input sources, such as consistent surface runoff, groundwater, or controlled sediment release, contribute to smaller fluctuations in total phosphorus concentration. Additionally, biological absorption and deposition processes play a role. Organisms present in the water, such as phytoplankton and algae, absorb total phosphorus and convert it into biomass. Moreover, some total phosphorus can also deposit into sediment. These processes help stabilize the concentration of total phosphorus and reduce fluctuations. Environmental conditions in the water, such as light intensity, temperature, and dissolved oxygen levels, can also influence total phosphorus concentration. When these conditions remain relatively stable without significant changes, the biological transformation and sedimentation processes related to total phosphorus also remain stable, leading to smaller fluctuations in total phosphorus concentration.

The concentration of Total Nitrogen (TN) remains relatively high from June to September each year. Several factors contribute to this observation. Firstly, the summer season is associated with increased nutrient inputs due to vigorous plant growth. Factors such as fertilization in farmlands and green spaces, irrigation in farmlands, and rainfall contribute to higher nutrient (including nitrogen) inputs into water bodies, resulting in relatively higher concentrations of total nitrogen during summer. Additionally, summer is characterized by abundant sunlight and higher temperatures, providing favorable conditions for the growth of algae and phytoplankton in the water. This enhances the mixing of nitrogen-rich water from the bottom layer with surface water, contributing to an increase in total nitrogen concentration.

### 3.2. Performance Comparison of Models

Based on the findings presented in Table 5 and Figure 5, the predictive performance of various machine learning models on Dissolved Oxygen (DO) content was evaluated, leading to a comparative analysis and discussion of each model’s performance in predicting water quality.

The Support Vector Machine (SVM) model achieved notable results with an $R^{2}$ score of 0.820, an MSE of 4.816, and an RMSE of 2.195 for DO prediction. These values indicate a relatively small average prediction error, showcasing the model’s ability to accurately predict the DO values. Moreover, the NSE value of 0.823 suggests that the model outperformed predictions based on the mean value alone. The SVM model’s proficiency in handling nonlinear relationships is crucial since DO levels are influenced by intricate nonlinear associations with multiple factors. By mapping the input space to a higher-dimensional feature space using a kernel function, the SVM model achieves improved fitting of nonlinear relationships. Additionally, the model’s decision boundary is determined by maximizing the margin, highlighting its strong generalization capabilities. This capability enables the SVM model to maintain good prediction performance when confronted with new samples, effectively avoiding overfitting or underfitting issues. Furthermore, the SVM model demonstrates resilience in the presence of noisy or outlier-laden data by selectively considering support vectors, thereby enhancing prediction accuracy. These characteristics collectively contributed to the model’s commendable performance in predicting the DO levels.

On the other hand, the Multilayer Perceptron (MLP) model exhibited slightly larger prediction errors compared to the SVM model. The MLP model achieved an $R^{2}$ score of 0.775, an MSE of 6.128, and an RMSE of 2.473 for the DO prediction. While the prediction error was slightly larger than that of the SVM model, the NSE value of 0.775 suggests that the predicted results were comparable to those obtained using the average value. The performance of the MLP model heavily relies on the design of its network structure. In cases where the network structure is not suitable, the model may struggle to capture the complex nonlinear relationship of DO content accurately. Moreover, the MLP model typically performs better when applied to large-scale and high-quality datasets. Insufficient training data or the presence of numerous noises or outliers may adversely affect the model’s performance. Unlike the SVM model, the MLP model involves numerous hyperparameters that require optimization, such as the number of hidden layer nodes, the selection of the activation function, and the learning rate. The improper selection or insufficient tuning of these hyperparameters can negatively impact the predictive performance of the model.

The Random Forest (RF) model demonstrated larger prediction errors compared to the SVM and MLP models, with an $R^{2}$ score of 0.720, an MSE of 7.613, and an RMSE of 2.759 for DO prediction. The NSE value of 0.720 indicates that the RF model’s predictions were slightly inferior to the baseline prediction using the mean value. The performance of the RF model in predicting DO levels can be influenced by the presence of class imbalance in the training data. Specifically, the distribution of DO levels showed an imbalance, with fewer samples in the 0–5 and 15–20 concentration ranges and more samples in the 5–15 concentration range. This imbalance can result in poorer performance of the RF model when predicting the DO levels within the underrepresented concentration ranges. Additionally, the performance of the RF model heavily relies on the number and depth of the decision trees. Opting for a small number of trees or shallow trees may hinder the model’s ability to capture the complex relationships in the DO levels, leading to larger prediction errors. Thus, it is essential to carefully select the number and depth of the decision trees to enhance the RF model’s predictive performance.

The XGBoost model exhibited larger prediction errors compared to the other models, with an $R^{2}$ score of 0.690, an MSE of 8.403, and an RMSE of 2.899 for DO prediction. In comparison to the other models, the prediction error was relatively high. The NSE value of 0.691 indicates that the XGBoost model’s predictions were comparable to the baseline prediction using the mean value. Similar to the Random Forest (RF) model, the performance of the XGBoost model can be influenced by class imbalance in the training data. If there is an imbalance in the distribution of the DO levels, with certain concentration ranges having fewer samples, it can impact the model’s ability to predict DO levels within those ranges. The XGBoost model possesses strong capabilities in capturing interactions and nonlinear relationships among features. When the variations in DO levels are complex and driven by intricate interactions or nonlinear relationships, the XGBoost model may require a larger number of trees or deeper trees to effectively capture these relationships and enhance the prediction performance. Consequently, the selection of an appropriate number and depth of the trees becomes crucial for achieving improved performance with the XGBoost model.

The LSTM model demonstrated high prediction accuracy for the DO values, with an $R^{2}$ score of 0.882, an MSE of 3.361, and an RMSE of 1.827. These values indicate that the LSTM model had a small average prediction error and performed with high accuracy in predicting the DO values. The NSE of 0.877 suggests that the model’s predictions were superior to those obtained using the average value. The LSTM model is a recurrent neural network model specifically designed for processing time series data. Given the time dependence of the DO content, the LSTM model can effectively capture the dynamic changes and trends, thereby improving the prediction accuracy. Through its gating mechanisms and memory units, the LSTM model is capable of retaining and updating important information while disregarding irrelevant information. This long-term memory capability enables the LSTM model to capture the long-term dependence of the DO content, resulting in improved prediction accuracy. Moreover, the LSTM model has the ability to automatically learn and extract features relevant to the DO content prediction. It can adaptively adjust the weight of features to maximize the extraction of useful information. This feature extraction capability contributes to the model’s enhanced predictive performance for the DO content. In summary, the LSTM model performed well in predicting the Dissolved Oxygen (DO) content due to its effective processing of time series data, long-term memory ability, feature extraction ability, and capacity to capture the sequential nature and patterns of the DO content. These factors collectively enable the LSTM model to achieve high prediction accuracy and demonstrate relatively good performance in DO content prediction.

### 3.3. Comprehensive Prediction Performance of the LSTM Model

The LSTM model’s architecture comprises of four LSTM layers, each consisting of 56 neurons. To mitigate overfitting, a dropout layer was added after each LSTM layer with a dropout rate of 0.2. The final layer of the model was a dense layer containing only one neuron, responsible for generating the prediction results. The input variable sequence length was set to 30, implying that the model uses water quality data from the previous 5 days to predict the water quality parameter for the subsequent time period. Parameter optimization for the model was performed using GridSearchCV, leading to the identification of the optimal parameter combination as follows: the batch size was 32; the epochs were 60; the optimizer was Adam.

Based on the findings presented in Table 6 and Figure 6, the LSTM model demonstrated exceptional predictive performance for key variables such as Dissolved Oxygen (DO), Ammonia Nitrogen (NH${}_{3}$-N), Total Nitrogen (TN), and Total Phosphorus (TP). Specifically, the $R^{2}$ values for these variables were 0.882, 0.830, 0.745, and 0.773, respectively. These values indicated a strong correlation between the predicted and actual values, highlighting the effectiveness of the LSTM model. Additionally, the corresponding MSE and RMSE values were relatively small, indicating low average prediction errors and the overall high performance of the LSTM model. The NSE values of 0.829, 0.745, and 0.763 further supported these results, surpassing the predictions based on the mean values for these variables.

However, it is noteworthy that Figure 6c reveals a relatively weaker predictive performance of the LSTM model for TN content, particularly in capturing extreme values accurately. This observation can be attributed to several contributing factors. Firstly, the complexity of the model may not be adequate to capture the intricate nonlinear relationships and long-term dependencies within the data, particularly when predicting extreme values. Therefore, it is recommended to consider utilizing a more-sophisticated model structure or enhancing the model’s capacity to improve its ability to predict extreme values. Secondly, improper feature selection could be another influential factor. Although a comprehensive weight method, along with the entropy weight method and Pearson correlation coefficient method, was employed for feature screening, the selection of individual feature variables may have been subjective. Specifically, when predicting TN content, the input features of the LSTM model may not effectively capture the characteristics necessary to handle extreme information. Thirdly, the presence of noise or outliers in the data can negatively impact the accurate prediction of extreme values by the model. Lastly, insufficient training could also contribute to the relatively weaker performance of the LSTM model. To enhance TN content prediction, it is advisable to increase the training sample size and extend the training duration to enable the model to fully capture extreme patterns. Insufficient training samples or a relatively short training period may hinder the model’s ability to effectively capture extreme features. To address these potential limitations, future improvements can include increasing the training sample size, adjusting the model structure, optimizing the feature selection, and conducting longer training sessions to enhance the overall predictive performance of the LSTM model for TN content.

The LSTM model’s proficiency in capturing dynamic features and trends in time series data significantly contributed to its accurate prediction of water quality variables. By leveraging its ability to learn temporal relationships and long-term dependencies within the sequence data, the LSTM model excelled in predicting future values of water quality variables. Furthermore, its capability to handle nonlinear relationships and complex temporal patterns further strengthened its performance in predicting water quality variables.

## 4. Conclusions

The primary objective of this study was to evaluate the predictive capabilities of various machine learning models for water quality parameters using the entropy weighting method. A comprehensive weighting method was proposed, which combines the entropy weighting method with the Pearson correlation coefficient method, for feature selection in water quality prediction. This method takes into account both the information entropy of the input features and their correlation with the target variable, effectively identifying the features possessing a significant impact on water quality variable prediction. The method offers valuable insights for subsequent water quality prediction modeling, including feature set selection, the reduction of redundant features, and the optimization of model performance.

Multiple machine learning models were investigated for their applicability in water quality prediction, the Support Vector Machine (SVM), Multilayer Perceptron (MLP), Random Forest (RF), XGBoost, and LSTM models. These models demonstrated varying capabilities in water quality prediction. SVM, in particular, exhibited good generalization performance and high prediction accuracy, specifically for the prediction of Dissolved Oxygen (DO). The MLP model, known for its strong nonlinear modeling capability, performed well in predicting DO and NH${}_{3}$-N, explaining a significant proportion of the target variable’s variance and exhibiting relatively small prediction errors.

In contrast, the RF model, despite its ability to handle high-dimensional data and complex relationships, showed relatively poor performance in water quality prediction. It displayed lower $R^{2}$ values and higher MSE and RMSE values, indicating larger prediction errors. This could be attributed to the model’s limitations in capturing complex relationships and extreme values in water quality data, leading to decreased prediction accuracy. Similarly, the XGBoost model also exhibited relatively poor predictive performance, with lower $R^{2}$ values and higher MSE and RMSE values, indicating larger prediction errors. This might be due to the model’s limited ability to capture complex relationship patterns and extreme values in the water quality data, resulting in lower prediction accuracy compared to the other models.

The LSTM model demonstrated excellent water quality prediction capabilities. As a recurrent neural network model designed to handle sequential data, LSTM possesses strong memory and long-term dependency modeling capabilities. In water quality prediction, the LSTM model effectively captured dynamic changes in time series data and consistently delivered outstanding predictive performance for various water quality parameters. Its high $R^{2}$ values and NSE values, along with low MSE and RMSE values, indicated small average prediction errors and significant improvements over simple mean value prediction methods.

In summary, the comprehensive weighting method that combines the entropy weighting method and the Pearson correlation coefficient method showed effectiveness in selecting a feature set for water quality prediction, enhancing the predictive performance of the models. Through comparative studies, the LSTM model emerged as the top-performing model for water quality prediction, accurately forecasting variations in different water quality variables in a stable manner. These research findings provide essential insights for water quality monitoring and management, assisting water quality management agencies in making informed decisions and devising effective management strategies. However, further research and applications are necessary to explore optimized feature-selection methods, improve machine learning models, and enhance the accuracy and reliability of water quality prediction.

## Figures and Tables

**Figure 1 entropy-25-01186-f001:**
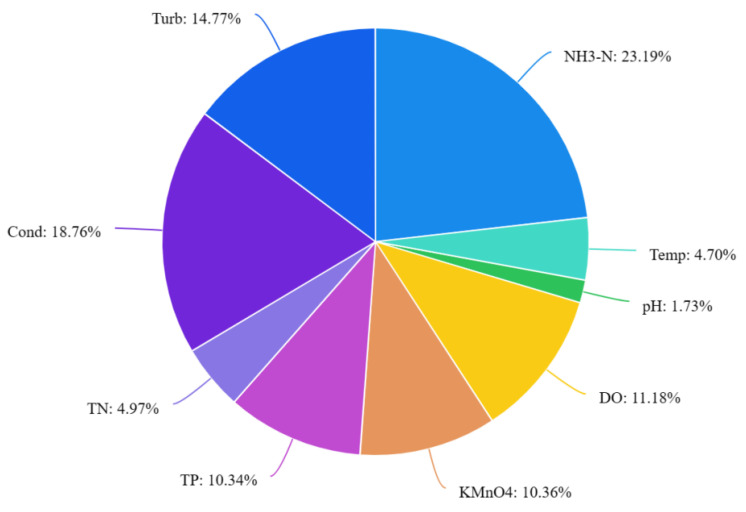
Information weight distribution map (according to Table 1).

**Figure 2 entropy-25-01186-f002:**
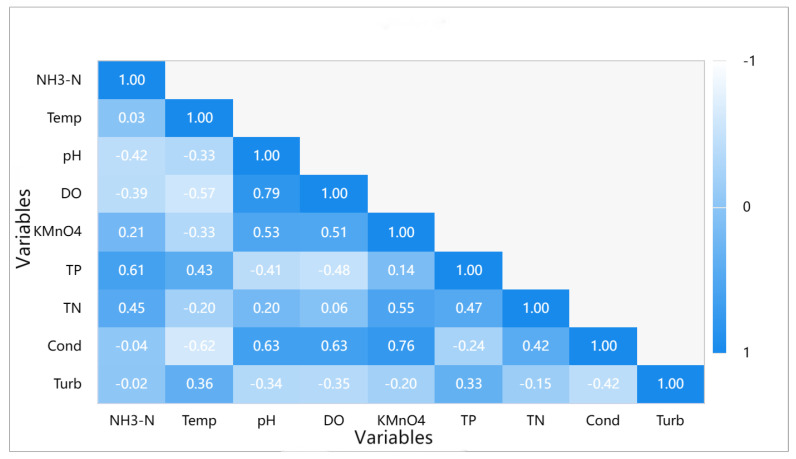
Visualization of Pearson correlation.

**Figure 3 entropy-25-01186-f003:**
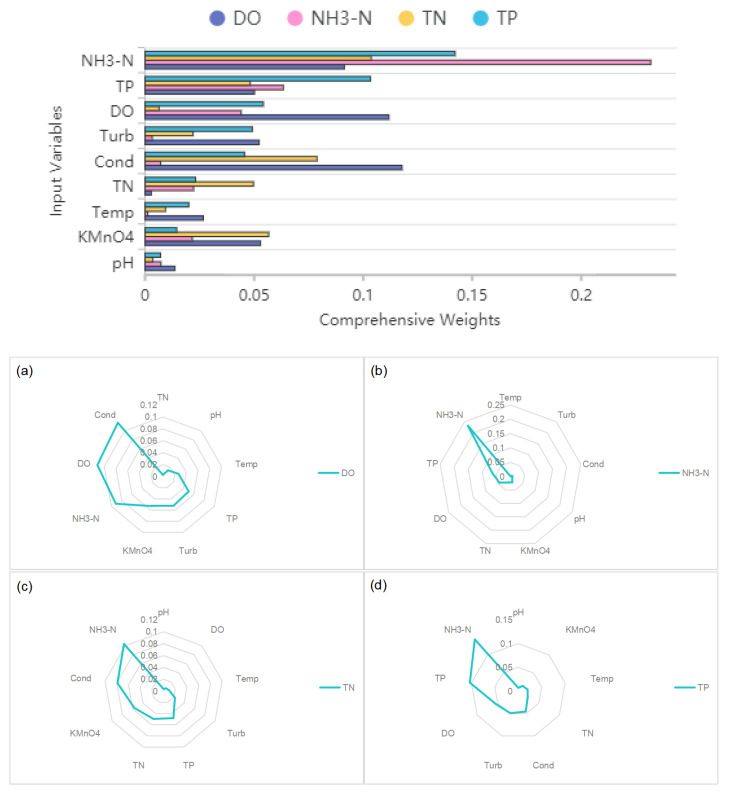
The comprehensive weight of input variables under each output variable: (**a**) DO and comprehensive weights of its corresponding input variables; (**b**) NH${}_{3}$-N and comprehensive weights of its corresponding input variables; (**c**) TN and comprehensive weights of its corresponding input variables; (**d**) TP and comprehensive weights of its corresponding input variables.

**Figure 4 entropy-25-01186-f004:**
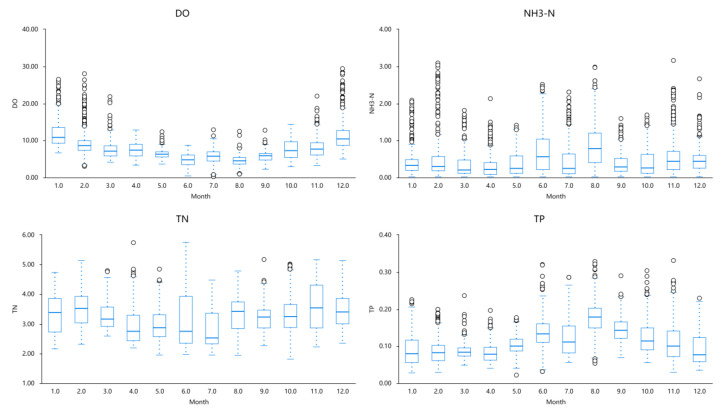
Fluctuationof water quality variables (The black circle in figure body represents Outlier).

**Figure 5 entropy-25-01186-f005:**
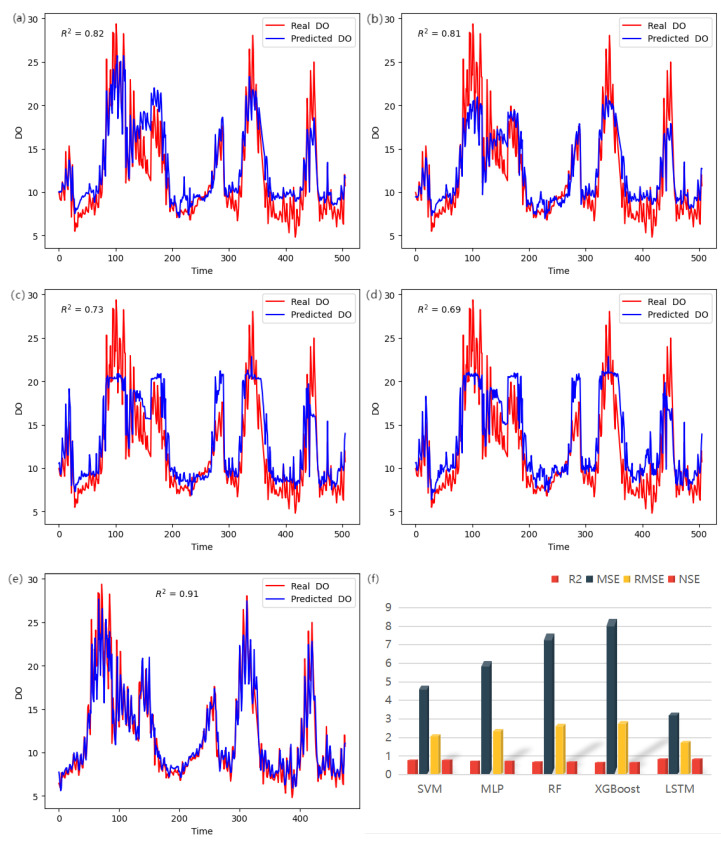
(**a**) Comparison of real values and predicted values for the SVM model; (**b**) Comparison of real values and predicted values for the MLP model; (**c**) Comparison of real values and predicted values for the RF model; (**d**) Comparison of real values and predicted values for the XGBoost model; (**e**) Comparison of real values and predicted values for the LSTM model; (**f**) Performance evaluation graph for each model.

**Figure 6 entropy-25-01186-f006:**
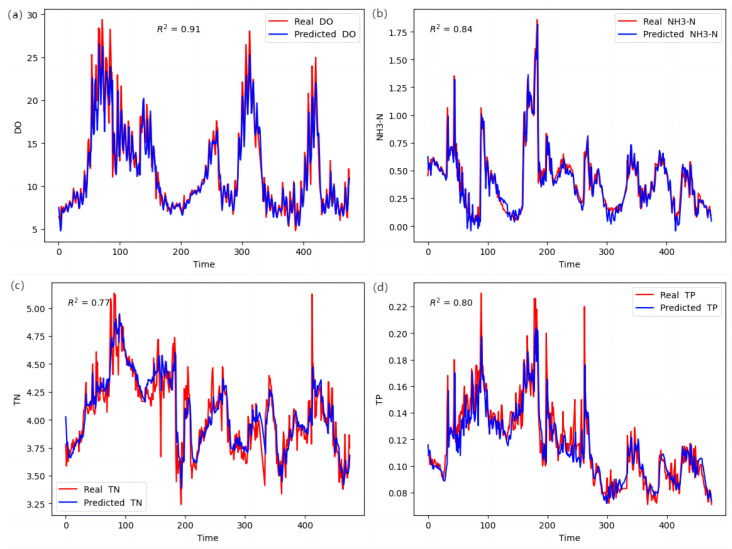
Comparison of LSTM model’s actual and predicted values for other water quality parameters: (**a**) Comparison of true and predicted values of DO; (**b**) Comparison of true and predicted values of NH${}_{3}$-N; (**c**) Comparison of true and predicted values of TN; (**d**) Comparison of true and predicted values of TP.

**Table 1 entropy-25-01186-t001:** Results of information weight calculation using entropy weight method.

Variables	Mean	Standard Deviation	$\mathrm{CV}\mathrm{Coefficient}{\;}^{1}$	$\mathrm{Weight}{\;}^{2}$
NH${}_{3}$-N	0.484	0.464	0.958	0.232
Temp	24.504	4.763	0.194	0.047
pH	7.554	0.540	0.072	0.017
DO	7.887	3.642	0.462	0.112
KMnO${}_{4}$	3.715	1.591	0.428	0.104
TP	0.110	0.047	0.427	0.103
TN	3.274	0.672	0.205	0.050
Cond	1264.521	979.968	0.775	0.188
Turb	49.761	30.356	0.610	0.148

^1^ CV coefficient = standard deviation/mean. ^2^ The weights are calculated by normalizing the CV coefficients.

**Table 2 entropy-25-01186-t002:** Summary of comprehensive weight results (The comprehensive weight values of different input variables for predicting the target variable).

Variables	DO	NH${}_{3}$-N	TN	TP
NH${}_{3}$-N	0.091	0.232	0.104	0.142
Temp	0.027	0.001	0.009	0.020
pH	0.014	0.007	0.003	0.007
DO	0.112	0.044	0.006	0.054
KMnO${}_{4}$	0.053	0.022	0.057	0.015
TP	0.050	0.063	0.048	0.103
TN	0.003	0.022	0.050	0.023
Cond	0.118	0.007	0.079	0.046
Turb	0.052	0.003	0.022	0.049

**Table 3 entropy-25-01186-t003:** Model evaluation methods.

Metric	Description	Formula
Mean-Squared Error (MSE)	The MSE measures the average difference between predicted values and true values in regression models. It is calculated as the mean of the squared differences between the predicted and true values.	MSE = (1/*n*) * Σ($y_{pred}$ − $y_{true}$)^2^
Root-Mean-Squared Error (RMSE)	The RMSE is the square root of the MSE and provides a measure of the average error between predicted and true values. It is consistent with the scale of the true values, making it easier to interpret.	RMSE = sqrt (MSE)
Nash–Sutcliffe Efficiency (NSE)	The NSE is a metric commonly used in hydrological models to evaluate the fit between model predictions and observed values. It considers the ratio of the sum of squared differences to the variance of observed values, subtracted from 1.	NSE = 1 − (Σ($y_{pred}$ − $y_{true}$)^2^/Σ($y_{true}$ − $y_{mean}$)^2^)
Coefficient of Determination (*R*^2^ Score)	The $R^{2}$ score evaluates the model’s ability to explain the variance in the observed data. It calculates the ratio of the sum of squared differences between predicted and observed values to the total variance of the observed values.	*R*^2^ Score = 1 − (Σ($y_{pred}$ − $y_{true}$)^2^/Σ($y_{true}$ − $y_{mean}$)^2^)

**Table 4 entropy-25-01186-t004:** Statistics of various water quality variables’ data.

Variables	Sample Size	Min	Max	Mean	Standard Deviation	Median
DO	5058	0.323	29.370	7.887	3.642	7.160
NH${}_{3}$-N	5058	0.025	3.162	0.484	0.464	0.352
TN	5058	1.811	5.755	3.274	0.672	3.243
TP	5058	0.022	0.331	0.110	0.047	0.101

**Table 5 entropy-25-01186-t005:** Performance evaluation of various models for dissolved oxygen prediction.

Models	$R^{2}$	MSE	RMSE	NSE
SVM	0.820	4.816	2.195	0.823
MLP	0.775	6.128	2.473	0.775
RF	0.720	7.613	2.759	0.720
XGBoost	0.690	8.403	2.899	0.691
LSTM	0.882	3.361	1.827	0.877

**Table 6 entropy-25-01186-t006:** Performance evaluation of the LSTM model in predicting other water quality variables.

Variables	$R^{2}$	MSE	RMSE	NSE
DO	0.882	3.361	1.827	0.877
NH${}_{3}$-N	0.830	5.614	2.330	0.829
TN	0.745	5.747	2.352	0.745
TP	0.773	5.683	2.332	0.763

## Data Availability

Data shall be provided by the corresponding authors upon special request.

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
