# Peer review of "Water Quality Prediction Based on Machine Learning and Comprehensive Weighting Methods"

_entropy, 2023, doi:10.3390/e25081186_

Round 1

Reviewer 1 Report

The authors submitted the manuscript "Water quality prediction based on machine learning and comprehensive weighting methods” where they presented machine learning techniques to predict water quality. The study is potentially interesting and maybe publishable after major revision with additional simulation data. Here are the comments for authors:

- Table 1 needs explanation, what do variables mean? From where the numbers are? What does CV coefficient stand for, it was not defined before? etc.

- Same for figure 1, from where are %, how are they calculated?

- On what date were the methods used? Figure 2 shows correlations, but up to this point I cannot find from where are the date.

- Table 2 without explanation

- Figure 3 is not clear, what do the authors want to show with this figure?

- Table 4, from where is the data?

- It is not clearly explained in methods how different methods were used for predictions.

- It is also not show how these methods can be used for future predictions and how these predictions are relevant.

Reviewer 2 Report

Dear authors,
nice paper. I just have a few remarks:
1. Please add the source of your data. You just state:"For this study,
historical monitoring data were collected," without further explanation.

2. Explain the sentence: 

Machine learning is a kind of data- driven mathods used for prediction and classification[9].

 I can interpret this in different ways, anyone makes a lot of sence to me.

3. Table 4 is not references in the text. Please add that.

4. Please explain in more detail, how you tested the performance. While the outcome is well explained, I can't really find detailed information about the input data. What, how many  time steps you perdict? What is the format of the input (how long is the sample from which you want to predict ) ... Please give all necessary details to make it easy for another person to repeat your work

Good luck

Reviewer 3 Report

In this study, water quality prediction model was reported on the basis of machine learning and comprehensive weighting methods. This manuscript has the potential for publication, but there are some issues that need to be addressed:

1.    What is the most important factor in the prediction model? How did it influence the results?

2.    The model needs to be compared with the other reported in the literatures.

3. Some literatures in terms of water quality detection and monitoring (Analytical Chemistry, 2022, 94: 10091-10100; 10.1002/lom3.10488; 10.1016/j.seppur.2022.121703) should be cited in the revised version.

GOOD
